# Epstein–Barr Virus Infection Is Associated with Elevated Hepcidin Levels

**DOI:** 10.3390/ijms24021630

**Published:** 2023-01-13

**Authors:** Ximena Duque, Eugenia Mendoza, Segundo Morán, Mayra C. Suárez-Arriaga, Abigail Morales-Sánchez, José I. Fontes-Lemus, Diana A. Domínguez-Martínez, Ezequiel M. Fuentes-Pananá

**Affiliations:** 1Infectious Diseases Research Unit, Mexican Institute of Social Security, Mexico City 06725, Mexico; 2Gastroenterology Research Laboratory, Mexican Institute of Social Security, Mexico City 06725, Mexico; 3Research Unit on Virology and Cancer, Children’s Hospital of Mexico Federico Gómez, Mexico City 06720, Mexico

**Keywords:** Epstein–Barr virus (EBV), *Helicobacter pylori*, hepcidin, acute-phase proteins, iron deficiency anemia (IDA), bioinformatic analysis, gastric cancer, TCGA database

## Abstract

EBV and *Helicobacter pylori* (*H. pylori*) cause highly prevalent persistent infections as early as in childhood. Both pathogens are associated with gastric carcinogenesis. *H. pylori* interferes with iron metabolism, enhancing the synthesis of acute-phase proteins hepcidin, C-reactive protein (CRP), and α-1 glycoprotein (AGP), but we do not know whether EBV does the same. In this study, we correlated the EBV antibody levels and the serum levels of hepcidin, CRP, and AGP in 145 children from boarding schools in Mexico City. We found that children IgG positive to EBV antigens (VCA, EBNA1, and EA) presented hepcidin, AGP, and CRP levels higher than uninfected children. Hepcidin and AGP remained high in children solely infected with EBV, while CRP was only significantly high in coinfected children. We observed positive correlations between hepcidin and EBV IgG antibodies (*p* < 0.5). Using the TCGA gastric cancer database, we also observed an association between EBV and hepcidin upregulation. The TCGA database also allowed us to analyze the two important pathways controlling hepcidin expression, BMP–SMAD and IL-1β/IL-6. We observed only the IL-1β/IL-6-dependent inflammatory pathway being significantly associated with EBV infection. We showed here for the first time an association between EBV and enhanced levels of hepcidin. Further studies should consider EBV when evaluating iron metabolism and anemia, and whether in the long run this is an important mechanism of undernourishment and EBV gastric carcinogenesis.

## 1. Introduction

*Helicobacter pylori* (*H. pylori*) and Epstein–Barr virus (EBV) are two pathogens that cause life-long persistent infections that are highly common in children, particularly in developing countries. A global prevalence of 32.8% has been reported for *H. pylori* in children under 18 years of age [1], and a prevalence greater than 50% for EBV by the age of 6 years in the USA [2]. Both are considered as human carcinogens by the International Agency for Research on Cancer (IARC) [3]. *H. pylori* colonizes the gastric mucosa, and it is considered the main culprit of the local chronic inflammation leading to severe damage of the gastric mucosa. *H. pylori*-induced atrophic gastritis and intestinal metaplasia are considered important risk factors for progression to gastric cancer [4,5]. On the contrary, EBV is a lymphotropic virus that persists in B-lymphocytes and that has been associated with several types of B-cell lymphomas. However, EBV has also been associated with the development of carcinomas, more recently, also including gastric cancer [6,7].

EBV belongs to the family of the Herpesviridae, which are large enveloped viruses with double-stranded DNA that oscillate between latent and lytic infection states. EBV is transmitted via saliva droplets infecting the B-lymphocytes present in the lymph nodes of the Waldeyer’s ring, in which the virus establishes latent infections. Several of the viral latent antigens exhibit oncogenic activity, and latency is the state in which the virus is found in associated neoplasms [8,9,10,11]. The switch to the lytic viral cycle is called viral reactivation and it is characterized by expression of about one hundred viral genes, including transcription factors, polymerases responsible for replicating the viral genome, and structural proteins [12]. Infected individuals remain as EBV carriers for the rest of their life, usually without clinical symptoms, and the capacity of EBV to transit between latent and reactivation states facilitates this persistency. The contribution of the lytic cycle to viral carcinogenesis is not clear, but antibodies against structural proteins, such as the viral capsid protein (VCA), have been shown to correlate with nasopharyngeal carcinoma and gastric cancer [13,14]. Apart from immunodeficiency, there are no known states in which the virus–host balance is disrupted leading to carcinogenesis, such as in gastric cancer [15].

The gastric juice produced in the stomach is important for absorbing micronutrients [16]. *H. pylori* infection is also one of the etiological agents of iron deficiency (ID) and iron deficiency anemia (IDA) in children [17,18]. A recent meta-analysis showed that *H. pylori* infection increases the risk of IDA with a 2.2 odds ratio (95% confidence interval: 1.5–3.2) [17]. Indeed, IDA recovery has been demonstrated with successful *H. pylori* eradication and without involving iron supplementation [19], and IDA has been reproduced in murine models of *H. pylori* infection [20]. *H. pylori* interferes with iron absorption through multiple mechanisms, for instance eroding the gastric mucosa and altering the production of hydrochloric acid [21], but also by modulating the levels of proteins involved in iron metabolism, such as hepcidin [22,23]. Hepcidin is a peptide hormone synthesized in the liver. It participates in iron homeostasis by regulating the absorption of iron from the diet, and/or by controlling the recirculation of iron to and from iron stores through the hepcidin/ferroportin system [24,25]. Hepcidin is also a type II acute-phase peptide, whose synthesis increases in response to increased production of IL-6 and IL-1β during infectious-inflammatory processes [26,27]. During infection episodes, hepcidin levels rise together with other markers of inflammation, such as α-1 glycoprotein (AGP) and C-reactive protein (CRP). AGP is also considered an acute-phase protein, however, unlike CRP, its blood levels are very stable and remain high for a long time, even in cases of resolved acute infection, and for this reason it is also considered as a marker of chronic inflammation [28].

Contrary to *H. pylori*, the mechanism by which EBV triggers gastric cancer is not understood. Direct infection and transformation aided by expression of viral oncogenes is the main documented mechanism of EBV carcinogenesis [29]. Indeed, EBV-induced cancers usually show monoclonal viral infection in all or most tumor cells, including the EBV-induced gastric cancer [30,31,32]. Although there is some evidence that EBV is also implicated in the development of adult and pediatric gastritis and of gastric pre-neoplastic inflammatory lesions [14,33,34,35,36], today, it is not clear whether EBV infection also causes inflammation-dependent gastric damage and participates in the development of early and late gastric inflammatory lesions.

Multiple lines of evidence support interactions and synergy between *H. pylori* and EBV infection in the gastric mucosa [14,33,36,37,38,39,40,41,42]. These studies support that *H. pylori* facilitates the arrival of EBV-infected B-lymphocytes to the stomach [42], EBV reactivation and epithelial cell infection [40,41,43], triggering of severe lesions in coinfected individuals compared with those infected by only one pathogen [14,33,36,44], and synergism in their oncogenic activity [38]. Whether EBV infection also influences the levels of proteins related to iron metabolism, specifically hepcidin, has never been reported. The main goal of this study was to evaluate a potential link between EBV and hepcidin and the acute-phase proteins CRP and AGP. We studied two cohorts for this analysis, children from boarding schools in Mexico City and adults with gastric cancer from The Cancer Genome Atlas (TCGA) consortium. The former allowed us to also analyze the nutritional status, while the latter confirmed the observations made in children and allowed us to look for molecular mechanisms of EBV and hepcidin interaction. We observed a positive correlation between EBV antibodies and serum hepcidin levels (*p* < 0.5) in schoolchildren, and we confirmed this association in the transcriptional and genomic TCGA database. Furthermore, we linked the EBV–hepcidin association with an IL-1β/IL-6-dependent inflammatory pathway.

## 2. Results

*H. pylori* infection has been associated with iron deficiency (ID) and iron deficiency anemia (IDA) in children [17,18] through mechanisms of infection-induced gastric erosion and promotion of enhanced levels of hepcidin [22,23]. Because some lines of evidence support that EBV cooperates with the bacterium to trigger pediatric gastric inflammatory lesions [33,35], we aimed to study whether EBV could also be involved in iron homeostasis. The study included 145 schoolchildren from boarding schools in Mexico City; the mean age was 9.3 ± 1.7 years, 48.3% were girls, 33.8% were overweight or obese, and 5.5% were short for their age. A total of 15.9% had iron deficiency (ID) determined by low ferritin concentration; 16.2% presented biochemical data indicating the presence of an inflammatory process (elevated CRP and/or AGP). The median hepcidin concentration was 7.6 ng/mL (with values at the 25th and 75th percentiles of 3.6 ng/mL and 13.6 ng/mL, respectively). See Table 1 and Appendix A for all the anthropometric and biochemical measurements.

Table 2 shows the frequencies of *H. pylori* and EBV infection. A total of 34.5% (50/145) of schoolchildren had evidence of *H. pylori* infection, of which 48% (24/50) had active infection (^13^C-labeled urea breath test positive), and 52% had evidence of past infection (IgG antibodies against total extract of *H. pylori* and/or against CagA antigen positive). Of the schoolchildren with active or past *H. pylori* infection, 58% (29/50) were CagA positive. Regarding EBV, only 8.3% (12/145) of the children did not present antibodies (VCA-IgG, EA-IgG or EBNA1-IgG) for EBV infection, 91.7% were positive for one or more of the antibodies against EBV antigens, all of them positive for VCA-IgG indicative of past infection, or that the children were EBV carriers. There were two schoolchildren with VCA-IgM indicative of primary infection. For one of these children there was no serum for the biochemical measurements, and the other was eliminated from the study and unaccounted in the final analysis. A total of 32.4% (47/145) of the children presented evidence of *H. pylori* and EBV coinfection, 2.1% (***N*** = 3) of the children had only *H. pylori* infection, 59.1% (***N*** = 86) had only EBV infection, and 6.2% (***N*** = 9) were negative for both pathogens.

Table 3 shows the serum concentrations of ferritin, hepcidin, CRP, and AGP according to the infection status of EBV alone and in coinfection with *H. pylori*. The relationship between hepcidin and IDA with *H. pylori* has been previously documented [46,47,48]. Remarkably, we observed that the serum levels of hepcidin, CRP, and AGP were significantly higher in schoolchildren with EBV infection than in those without EBV infection (see also Figure 1). The ferritin concentration was not different. Hepcidin, CRP, and AGP levels were higher in EBV carrier children, and/or with *H. pylori* infection and/or in children with an inflammatory process, than in children non-infected or without an inflammatory process. Figure 2 shows that all the markers measured of EBV infection: VCA-IgG (R = 0.302, *p* < 0.001), EBNA1-IgG (R = 0.244, *p* = 0.011), and EA-IgG (R = 0.345, *p* = 0.011) had a positive correlation with the hepcidin concentration. We did not see significant correlations between EBV antibodies and CRP and AGP. The only other positive correlation was between AGP and hepcidin (R = 0.188, *p* = 0.047; not shown).

When we included the coinfection with *H. pylori* in the analysis, we observed that hepcidin and AGP remained associated with EBV infection irrespective of *H. pylori* infection, while CRP seemed dependent on infection with both pathogens. For hepcidin, the median (ng/mL) was coinfection = 8.9, EBV alone = 8.6, *H. pylori* alone = 1.5; the AGP median (g/L) was coinfection = 0.66, EBV alone = 0.68, *H. pylori* alone = 0.31; and the CRP median (mg/L) was coinfection = 0.32, EBV alone = 0, *H. pylori* alone = 0. A multivariate linear regression analysis showed similar results to those obtained with the univariate analysis; EBV infection was independently associated with hepcidin concentration, no association was observed between hepcidin and *H. pylori* infection, nor was any significant interaction or effect modification identified by the presence of *H. pylori*. Regardless of these associations between EBV and hepcidin, CRP and AGP, we did not observe a significant association between EBV infection and overweight/obesity: EBV positive = 35.9% (47/131) vs. EBV negative = 16.7% (2/12), stunting: EBV positive = 6.1% (8/131) vs. EBV negative = 0% (0/12), or iron deficiency (ID): EBV positive = 15.3% (20/131) vs. EBV negative = 16.7% (2/12). We also did not observe an association of these clinical parameters with any of the infections or with inflammation, perhaps due to the small size of the negative group for these three variables (***N*** = 7). See Appendix A for the correlation between the nutritional and infection status.

Because of the evidence supporting that EBV infection influences the development of gastric lesions in cooperation with *H. pylori*, and that *H. pylori* affects hepcidin levels and iron concentration through erosion of the gastric mucosa [14,21,22,23,33,35,36,37,38,39,40,41], we decided to evaluate the expression of *HAMP* (encodes for hepcidin), *A1BG* (encodes for AGP), and *CRP* (encodes for CRP) in the transcriptomic and genomic database of gastric cancer of the TCGA consortium (***N*** = 250). We also used the well-established separation of samples in those associated with EBV infection (EBVaGC; ***N*** = 24) and those negative to EBV (EBVnGC; ***N*** = 226) [6]. Using this database, we aimed to validate whether there was a correlation between EBV infection and the expression of hepcidin and the other type II acute-phase proteins. We first used the z-score to compare the expression of the genes of interest between EBVaGC vs. EBVnGC samples. Interestingly, we observed that, as in children, the EBVaGC samples have elevated *HAMP* expression (*p* = 0.0018), while we did not observe significant associations with the expression of *A1BG* (*p* = 0.1486) and *CRP* (*p* = 0.1534) (Figure 3A and Appendix A).

The association between EBVaGC and hepcidin (*HAMP*) led us to continue the analyses in this cohort of patients, aiming to identify the link between EBV and the regulatory mechanisms involved in the expression of *HAMP*. Previous studies have shown that hepcidin expression is regulated by at least two main pathways: the BMP–SMAD pathway that operates in response to high iron concentrations, and the JAK/STAT pathway that is active during acute and chronic inflammatory processes. We listed all the genes reported operating in these pathways and constructed a protein–protein interaction network (PPIN) to determine the involvement of each of the selected genes in the signaling pathways that regulate hepcidin expression. To perform this analysis, we used a list of genes documented in previous studies that includes ligands, receptors, modulators, and signaling intermediates of ligand–receptor pairs (see Appendix A for the list of the relevant genes and their references). The PP1N of Figure 3B clearly shows the separation of both pathways, with *HAMP* remaining as central to both pathways. The constructed PPIN consisted of 24 nodes and 104 edges, with one cluster harboring the elements of a BMP–SMAD signature, and the other harboring elements of an inflammatory signature in which *IL-1β, IL-6* and *IL-22* cytokines were present. Moreover, within the main biological processes and molecular functions associated with the PPIN network, we found hepatic immune response, response to iron ion starvation, transferrin receptor activity, and BMP receptor activity, among others (Figure 3C).

The PPIN and (GO) analysis strongly supported the signatures representing both hepcidin regulatory mechanisms. Therefore, we decided to evaluate the participation of both mechanisms in the regulation of hepcidin expression in the transcriptomic data of the EBVaGC and EBVnGC patients. We performed a single-sample gene set enrichment analysis (ssGSEA) to assess the enrichment of both signatures in both groups, finding a positive association between the inflammatory arm of the hepcidin regulation and the EBVaGC (*p* = 0.0007) (Figure 3D). On the contrary, the signature for the BMP–SMAD regulatory element was associated with the EBVnGC samples (*p* = 0.0056). Since the inflammatory signature was associated with hepcidin regulation in EBVaGC, we performed a centrality analysis to assess which components of the signature were the most important. We found that the most important proteins are cytokines IL-6 and IL-1β and their signaling intermediaries JAK2 and STAT3 (Figure 3E).

Finally, we decided to evaluate whether the inflammatory mechanism regulating the expression of hepcidin was a random process, mainly given by genetic alterations in which EBV may just encounter a more favorable environment. We generated a heatmap of single nucleotide variants, such as non-sense mutations, truncation mutation (either putative driver or of unknown significance), missense mutation (putative driver), and mutations in splicing sequences of unknown significance; and copy number alterations (deletions and amplifications). We did not find an enrichment of mutations in hepcidin or in the hepcidin-regulatory inflammatory pathway in the EBVaGC patients explaining its elevated expression in this group (Appendix A). Although this analysis does not allow us to conclude that EBV is the driving force activating the inflammatory pathway and the enhanced expression of hepcidin, it invites further research on this topic, placing the attention on EBV infection as a potential element influencing hepcidin levels, iron metabolism, and anemia in children and adults, and whether in the long run this is an important mechanism of undernourishment and EBV gastric carcinogenesis.

## 3. Discussion

In this study, we document an association between EBV infection and increased levels of hepcidin, both in schoolchildren and adults with EBVaGC. To the best of our knowledge, this is the first evidence placing EBV as a potential modulator of iron metabolism/homeostasis. Hepcidin is a peptide hormone that is essential to maintain systemic iron homeostasis. Iron is an indispensable element for humans and microorganisms, and iron chelation is a host mechanism to attenuate infection [49,50,51]. Many bacteria have developed strategies to counteract this host antagonistic mechanism and to facilitate iron accumulation. We and other groups have previously shown that *H. pylori* infection is associated with elevated hepcidin levels and with ID or IDA [17,19,20,22,23,46,52]. However, there are many causes of anemia, including iron deficiency, deficiencies of other minerals and vitamins and non-nutritional factors, such as acute and chronic inflammatory and infectious processes [53,54,55].

Viruses depend on iron to replicate in their host cells given that iron is essential for fundamental cellular processes, such as ATP and nucleic acids synthesis, and certain steps of the replication cycle of some viruses can be iron-dependent or iron-regulated [56]. It is known that some viral proteins can disturb iron homeostasis, for example US2, a protein encoded by HCMV (human cytomegalovirus). US2 regulates iron metabolism by targeting the haemochromatosis-associated protein HFE (homeostatic iron regulator protein, also known as High FE^2^+) for its proteasomal degradation, resulting in the accumulation of iron in the infected cell, which may be an advantage for the replication of larger viruses [57]. Iron serum and ferritin levels are elevated in patients with chronic hepatitis associated with hepatitis B virus (HBV) infection. A proposed mechanism explaining the increased iron release from hepatocytes is the liver damage induced by the infection [58]. In chronic hepatitis patients with elevated ferritin levels, there is a positive feedback where hepcidin is upregulated in the liver in response to elevated iron stores [59]. There is also evidence that HIV-1 (human immunodeficiency virus 1) infection correlates with high levels of intracellular iron and hepcidin, and that this enables the virus to increase viral gene transcription. Treatment with an iron chelator leads to downregulation of NFκB activity, which correlates with a decrease in HIV-1 reactivation [60]. In general, it is thought that the virally induced increase in intra-cellular iron ensures efficient viral replication, and iron overload is associated with poor disease prognosis. This effect has been observed at least during HBV, HCV (hepatitis C virus), HIV-1, and HCMV infections [61]. In fact, the pandemic SARS-CoV-2 (severe acute respiratory syndrome coronavirus 2) encodes its own hepcidin homologous protein, the covidin protein [62].

The association between EBV and hepcidin suggests that EBV may also be involved in modulating iron metabolism. Little is known about EBV or any other virus and iron metabolism, beside the above-mentioned examples. Altogether, these observations have been used to propose that interfering with iron homeostasis may be useful to treat chronic viral infections. Indeed, we observed a better correlation between hepcidin and antibodies against lytic markers VCA and EA than with antibodies against the latent protein EBNA1. The association between EBV antibodies directed against lytic proteins and gastric and nasopharingeal carcinoma has led us to propose that viral reactivation from latently infected B-cells is an essential step for epithelial cell infection and carcinogenesis [14,63,64,65]. Also, it has recently been reported that in EBV-positive epithelial cancer cells, iron chelation promotes autophagy and viral reactivation [66]. In this scenario, EBV may be altering iron homeostasis in order to support the viral replication in B-cells that facilitates epithelial cell infection. However, further research is needed to assess a relationship between iron homeostasis and EBV infection.

Since the association between EBV and hepcidin (*HAMP*) was also observed in EBVaGC, we performed a transcriptional analysis directed to identify the regulatory elements of *HAMP* expression in gastric cancer. Although this analysis was initiated with elements already known to regulate hepcidin synthesis, it was agnostic to EBV infection and gastric cancer. We found two clearly separated regulatory pathways active in gastric cancer, identified as BMP–SMAD and inflammatory. EBV was significantly associated with the inflammatory pathway, in which cytokines IL-1β and IL-6 seemed to be the most important drivers. It is considered that the synthesis of hepcidin in the liver is regulated by IL-1β and IL-6 [26], and that this synthesis increases in response to acute or chronic inflammatory or infectious processes [67,68]. In addition to hepcidin, we also observed elevated levels of CRP and AGP in schoolchildren carriers of EBV infection, which suggest that EBV enhances the synthesis of hepcidin through mechanisms related to triggering type II acute-phase inflammatory responses. Altogether, these data support a mechanism of EBV-induced inflammation that results in elevated levels of type II acute-phase proteins, including hepcidin. Of these three proteins, hepcidin was the most consistently associated with EBV, in children and adults, and with all the EBV antibodies measured.

Both IL-6 and IL-1β control the expression of many acute-phase proteins during acute and chronic infection episodes. *HAMP*, *CRP,* and *AGP* expression is mediated through the activation of STAT3, C/EBP family members, and Rel proteins (NFκB), and these transcription factors are downstream of many inflammatory pathways [69]. *CRP* expression increases within hours after *Streptococcus pneumonie* and Influenza A virus infection [69,70]. IL-6 regulates *HAMP* expression during Influenza virus (PR8) or *Streptococcus pneumoniae* infections in mice and in human hepatocytes [71]. Infection with norovirus also increases *CRP*, *AGP*, and *HAMP,* and promotes changes in levels of micronutrients iron and vitamin A [72]. *HAMP* is upregulated (20- to 40-fold) in murine myeloid cells by TLR4 after recognition of LPS derived from *Pseudomonas aeuroginosa* or Group A *Streptococcus* [73], and up to 8-fold by TLR5 in peripheral blood mononuclear cells after recognition of flagellin from *Salmonella typhimurium* [74], murine Influenza A H1N1 and *Candida albicans* [74]. It has also been observed that during COVID-19 infection, there is an increase in serum ferritin caused by the cytokine storms present during severe disease [75].

There is also evidence of an increase in IL-6 and IL-1β secretion in EBV-associated diseases, such as infectious mononucleosis and hemophagocytic lymphohistiocytosis [76,77]. We have also reported an association between IL-1β, EBV, and graft-versus-host disease in immunosuppressed children with allogeneic hematopoietic progenitor cell transplantation [78]. Elevated levels of IL-1β and IL-6 have also been observed in patients with multiple sclerosis [79,80]. Peripheral blood cells taken from multiple sclerosis patients respond by secreting these cytokines upon stimulation with EBV antigens [76,81]. An in vitro study supports an upregulation of *IL6* expression upon EBV infection [76]. RTA, an EBV immediate early transactivator that controls viral reactivation, has been found to drive transcription of the *IL6* gene in cell lines derived from nasopharyngeal cancer [82]. IL-1β is also elevated in the EBV-positive nasopharyngeal carcinoma [83]. Still, in future studies, it is necessary to dissect the exact mechanism(s) by which EBV increases hepcidin levels and whether this provides an advantage for virus persistence (latency) or reactivation.

Although EBV mostly causes subclinical asymptomatic infections, it is important to note that our study cohort was taken from boarding schools serving particularly underprivileged children. These were children whose nutritional status or history of infection could be compromised, favoring states of EBV dysbiosis. Indeed, the frequency of EBV and *H. pylori* infection were high in this cohort. We have previously evaluated EBV serology in children, finding infection frequencies of 69.8% (median = 10 years old), and of 64.3% (median = 10.1 years old) [33,84], while in this cohort it was of 91.7% (median = 9.3 years old). We were surprised by the observed high prevalence of EBV infection in these children, and unfortunately, the low number of uninfected children probably hampered the statistical analysis to find other potential associations.

Because of the link between hepcidin and acute response proteins with iron metabolism, elevated hepcidin has been proposed to lead to hypoferremia and a lower response to iron supplementation [25,85]. One of the relevant warning points of this study is whether EBV infection could also contribute to the development of iron deficiency (ID) and/or anemia, since increased hepcidin synthesis is related to decreased iron absorption. We did not observe a relationship between EBV and any anthropometric or iron deficiency measurement; this may be because of the small number of children included, or that a longer follow-up is needed. A role of EBV in ID or anemia has not been previously studied. More studies are needed considering the high prevalence of EBV infection, particularly in very young children in developing countries.

### Study Limitations

The study has several limitations. First, although the original study from which the samples were taken included 350 schoolchildren, as we were increasing the number of tests, the biological material was consumed, ending with only *N* = 145 samples in which we were able to perform the reported tests. Looking for alternatives to interrogate an association between EBV and hepcidin upregulation, we decided to analyze the TCGA cohort of adults with gastric cancer as a proof of concept of the observations made in the cohort of schoolchildren. A more robust and longitudinal study is needed to confirm and expand the associations found in this study. Second, this is an observational study in which we report statistical associations. Although we observed significant (*p* ≤ 0.05) correlations between hepcidin and all the EBV antibodies, these correlations are low. In future studies, it will be important to correlate markers of EBV infection and iron metabolism in children in which iron is supplemented and *H. pylori* is pharmacologically eradicated, particularly if the children show anthropometric evidence of undernourishment and/or of chronic dyspepsia. Because of the socioeconomic status of the study cohort, multiple infections may be at play and affecting the nutritional status of the children. We would like to emphasize that the children were healthy at the time of sample collection. When the samples were collected, a questionnaire was applied to the children, and those with a recent symptomatic gastro, respiratory, or urinary infection were excluded from the study, or were programmed to take their samples weeks after resolution of the clinical symptoms. Moreover, the significant associations that we are reporting in this study were made by comparing children within the same cohort.

## 4. Materials and Methods

### 4.1. Study Population

This study was approved by the Ethical, Biosafety, and Scientific Institutional Review Boards of the Children’s Hospital of Mexico “Federico Gómez” (HIM 2016-002). The study involved children from 6 to 13 years of age, who initially participated in a longitudinal study about the effect of *H. pylori* infection on growth velocity in schoolchildren with low socioeconomical status. The children attend different boarding schools across Mexico City where they receive a homogeneous diet [86]. The children stay at school from Monday to Friday and return home on weekends and holidays. The children and their parents or tutors signed an informed consent form, and the original study was also authorized by the Mexican Social Security Institute (R-2014-785-023) and the Secretary of Public Education.

### 4.2. Sample Collection

The status of *H. pylori* infection was determined by a ^13^C-urea breath test and serologic assessment of whole anti-*H. pylori* and anti-CagA antibodies [86]. The ^13^C-urea breath test consisted of collecting two samples of expired air as explained below. Blood samples were also drawn on the same day in which the breath sample was collected. After 8 h of fasting, venous blood samples were collected in two tubes: one with Ethylenediamine tetraacetic acid (EDTA) to complete blood count, and another tube with a gel separator. The latter was centrifuged at 4000 rpm, sera was collected and aliquoted into cryovials, which were stored in an ultra-freezer at −80 °C. Different sera aliquots were thawed for the serological and biochemical determinations, and were not subjected to thawing/freezing cycles. The serum samples were kept frozen for eight years on average before being used to make the specific determinations of this study: the EBV serology, hepcidin, AGP, and CRP.

### 4.3. Determination of H. pylori Infection

The status of *H. pylori* infection was obtained from the main study. The basal sample for the ^13^C-urea breath test was obtained 10 min after the child had ingested a beverage containing 2 g of citric acid (Citra-LP; San Miguel Proyectos Agropecuarios S.P.R., Huichapan, Hidalgo, Mexico) to delay gastric emptying. Immediately after, children were given 50 mg of ^13^C-labeled urea dissolved in 150 mL of water, and the final sample was collected 30 min later. Expired air samples were collected in 10 mL tubes (Exatainers, Labco Ltd., High Wycombe, UK). A difference of ≥5 parts/1000 between ratio values ^13^CO_2_/CO_2_ at baseline and 30 min post-intake of ^13^C-urea was considered a positive test for active *H. pylori* infection. The tubes with expired air were stored at room temperature and analyzed by a mass spectrometer (BreathMat-plus, Finnigan MAT, Bremen, Germany). The sensitivity and specificity of this test is >90% in children aged 6 years or older [87,88,89,90].

Serological tests to IgG antibodies to whole *H. pylori* and CagA antigens were performed by ELISA and as previously reported [46,86]. Optical density ratio (ODR) values of ≥1.0 and ≥1.5 were considered positive for *H. pylori* and CagA, respectively. Whole *H. pylori* lysates and purified recombinant CagA were used as antigens. ODR values were calculated in relation to reference sera. These tests have been previously validated in Mexican pediatric populations [91]. The sensitivity and specificity of the tests are 85–87% for whole *H. pylori* and 83–97% for CagA [91].

### 4.4. Determination of Anti-EBV VCA Antibodies

Anti-EBV antibodies were also determined by ELISA specifically for this study, as it has been previously described [33,84]. Briefly, anti-EBV VCA antibodies were determined using the commercial kits (HUMAN; Wiesbaden, Germany) for IgG anti-VCA (catalog 51,204) and IgM anti-VCA (catalog 51,104) following manufacturer instructions. An amount of 100 µL of the appropriate patient-serum dilution (1:100 for IgM and 1:20 for IgG) was deposited on the corresponding well with the VCA antigen already attached, and incubated for 1 h (IgG) or 30 min (IgM) at 25 °C. After washing four times, 100 µL of peroxidase-conjugated anti-human IgG or IgM rabbit antibody were added and incubated for 30 min at 25 °C. The reaction was revealed with 100 µL of substrate reagent (3,3′,5,5′ tetramethylbenzidin (TMB)–hydrogen peroxide) and the plates were read in an ELISA reader (Thermo Scientific, multiskan ascent, Waltham, MA, USA) to an absorbance of 450 nm. The reported value is the average of two independent assays, while a subgroup of samples was assayed by quadruplicate using different lots of the ELISA kit to check for reproducibility. Assessment of the antibody titers was performed according to the manufacturer’s instructions and the values are reported as units/mL. Children were separated in two groups according to their EBV serostatus. The first group included non-infected children (EBV IgG and IgM negative), and the second group included children with an EBV persistent infection (IgG positive and IgM negative). Anti-EBV IgG antibodies against early antigen (EA) and nuclear antigen 1 (EBNA1) were also measured but they did not give significantly different results than those observed with anti-VCA IgG.

### 4.5. Determination of Hematological Parameters

Hematological and iron parameters were obtained from the main study, in which levels of hemoglobin (Hb) were determined by automated analyses using the cianometahemoglobin method (Coulter T-540). The hemoglobin concentration was adjusted by altitude [92]. Serum ferritin was determined by a radioimmunoassay (FER-IRON II, RAMCO Laboratories, Houston, TX, USA). Iron deficiency (ID) was defined as a ferritin < 15 ng/mL, anemia was defined as Hb < 11.5 g/dL for children aged less than 12 years, and Hb < 12.0 g/dL for children aged from 12 to 13 according to the WHO criteria [93]. IDA was defined as iron deficiency in combination with anemia, and non-IDA anemia as low Hb withoutID.

Hepcidin was measured in serum by competition ELISA, using the hepcidin-25 (human) enzyme immunoassay kit (S-1337; Bachem, San Carlos, CA, USA) with a detection range of 0–25 ng/mL, according to the manufacturer’s protocol. Standards were run in duplicate and single sample determination was done. Samples giving readings outside the linear region of the curve were diluted 1:4 in peptide-cleared human serum provided with the kit. Hepcidin-25 levels were calculated from a calibration curve with a linear measuring range near the IC50 of the curve 1.5 ng/mL (0.35–1.72). This assay used biotinylated hepcidin as competitor and synthetic hepcidin for calibration, the competitive format of the assay implied the inverse correlation between hepcidin concentration and the measured absorbance, which is often less sensitive than the direct correlation [94]. However, this assay has been proven to have good analytical precision compared with other hepcidin quantification methods [95,96]. Levels of AGP and CRP were determined in serum by ELISA (GWB-F43C70, GenWay Biotech. Inc. (San Diego, CA, USA) and RAP001 and Biovendor Research and Diagnostic Products (Asheville, NC, USA), respectively). Levels < 5 mg/L to CRP, and less than 1 g/L to AGP were considered as normal [97].

### 4.6. Anthropometric Indicators

For anthropometric data analysis, Z-scores of heights for age and body mass index (BMI) for age were obtained using WHO’s Anthro Plus software [98]. Children with height for age Z-score less than −2 standard deviations (SD) were classified with stunting and children with BMI for age Z-score higher than +1 SD were classified as overweight or obese [45].

### 4.7. TCGA Data Analysis

The expression of *HAMP* (encodes for hepcidin), *A1BG* (AGP), and *CRP* was analyzed in the gastric cancer database of TCGA [6] using the cBioPortal platform (https://www.cbioportal.org/, retrieved August 30, 2022). Gastric cancer samples were divided in those associated with EBV infection (EBVaGC; *N* = 24) and those negative to EBV (EBVnGC; *N* = 226), and the Z-score was used to compare the expression of each gene between both groups of samples.

To further understand the regulation of hepcidin and the potential mechanisms by which EBV affects hepcidin expression, a protein–protein interaction (PPIN) network was constructed using the online platform STRING (https://string-db.org, retrieved August 30, 2022). The proteins involved in regulation of hepcidin expression were taken from a review of literature (see Appendix A for a full list of genes involved and references). This list of genes was used as input to build the PPIN. The network was constructed with an interaction score of 0.700 (high confidence). To define each interaction, the following parameters were evaluated: neighborhood, gene fusion, co-occurrence, co-expression, experimental evidence, databases, and text mining.

Another PPIN was generated using the input of the inflammatory genes (*IL-1β, IL6ST, IL-22, IL-6R, STAT5B, STAT3, IL-6* and *JAK2*) and hepcidin (*HAMP*) in the platform STRING v11.5 (https://string-db.org, retrieved August 30, 2022). To define the interactions, we used the parameters described above. The interaction network obtained was imported into Cytoscape software v3.6.1 to perform a centrality analysis defining the importance of each of the nodes. Here, the size of the node represents the most central or most important proteins according to the betweenness centrality, with the largest nodes having the highest centrality.

The biological processes and molecular functions (GO) associated with the network were determined, identifying the terms with the highest FDR values resulting from the analysis platform. Subsequently, using the expression matrix of all the genes in all the included samples (*N* = 250), a ssGSEA was performed to assess how well the signatures of the two main regulatory mechanisms of hepcidin were represented in the EBVaGC and EBVnGC. The analysis was performed using the GSVA v1.30.0 package in R v3.5.1 with method = ‘ssgsea’. Normalized enrichment scores (NES) were calculated using the list established above as gene sets, which contains the main ligands, modulators, receptors, and signaling components of each mechanism (Appendix A). The analysis was performed on samples classified as EBVaGC and EBVnGC.

To know more about the potential mechanisms driven by the enrichment of the hepcidin signature, we evaluated genetic alterations in the list of relevant genes for hepcidin regulation. We also used the cBioPortal platform for this analysis. Heatmaps were created to denote the genetic alterations of the relevant genes identifying the samples as either EBVaGC or EBVnGC.

### 4.8. Statistical Analysis

The sociodemographic characteristics of the study are described using mean ± SD (standard deviation) or median and values at percentile 25 and 75 or percentages, according to distribution and type of variables. The frequency of infection by EBV was obtained from the different antibodies that were determined and is described as a percentage; the coinfection between EBV and *H. pylori* (obtained by ^13^C-urea breath test and by serologic tests) is also presented in percentages.

The median of acute-phase proteins: ferritin, CRP, AGP, and hepcidin between the groups according to the status of EBV infection, and by coinfection with *H. pylori* infection and the presence of EBV and/or *H. pylori* infection and/or inflammatory process were compared using the Mann–Whitney U test or Kruskal–Wallis test. If the Kruskal–Wallis test showed differences with statistical significance in the acute-phase proteins by groups of coinfection between EBV and *H. pylori*, Dunn with Bonferroni correction as a post-hoc test was used due to multiple comparisons. Spearman rank correlation coefficients were determined to assess relationships among EBV antibodies and ferritin, hepcidin, CRP, and AGP concentrations. We also compared the children’s nutritional status measurements by anthropometric indicators between the EBV infection or between the groups of children with EBV and/or *H. pylori* infection and/or inflammatory process, for which Fisher’s exact test was used. The association between EBV infection and hepcidin levels was also assessed using a multivariate linear model, which included an interaction term to assess whether *H. pylori* coinfection modifies the association between EBV infection and hepcidin levels. Statistical analyses were carried out using Statistical software Stata, version 12 (StataCorp, College Station, TX, USA).

The GraphPad Prism software v8.3.1. (GraphPad software, San Diego, CA, USA) was also used for the statistics of the bioinformatic analysis. For data with lacking normality and/or homogeneity, the Mann–Whitney–Wilcoxon test was used. Outliers were identified by the ROUT analysis method (Q = 1%) to Z-score comparison of *HAMP*, *A1BG,* and *CRP* genes. Significant p values were established at ≤0.05 (*), ≤0.01 (**), and ≤0.001 (***) in all statistical analyses.

## 5. Conclusions

The results of the study support that children who are carriers of EBV infection present high levels of hepcidin and other acute-phase proteins, such as CRP and AGP. Hepcidin and AGP remained high in children solely infected with EBV, while CRP was only significantly high in EBV and *H. pylori* coinfected children. The association between EBV and hepcidin was observed in schoolchildren, and also in adults with EBVaGC. The EBV influence on hepcidin expression may be a side effect of the virus-induced inflammation, since we also observed a link between IL-6 and IL-1β and EBVaGC, or a direct need of iron for the EBV biological cycle. However, regardless of whether this is a collateral or direct effect, EBV still may influence iron metabolism, and further studies should consider EBV infection when evaluating the nutritional status of children and adults, such as in ID, IDA, or anemia, and whether in the long run this is an important mechanism of EBV gastric carcinogenesis.

## Figures and Tables

**Figure 1 ijms-24-01630-f001:**
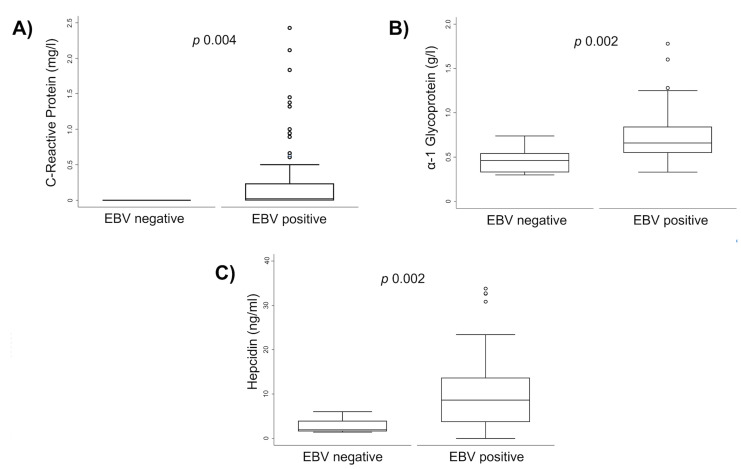
EBV infection is associated with enhanced levels of hepcidin and acute-phase protein in schoolchildren. Concentration of (**A**) C-Reactive protein, (**B**) α-1 Glycoprotein, and (**C**) hepcidin were measured in children and compared according to their EBV infection status using the Mann–Whitney’s test.

**Figure 2 ijms-24-01630-f002:**
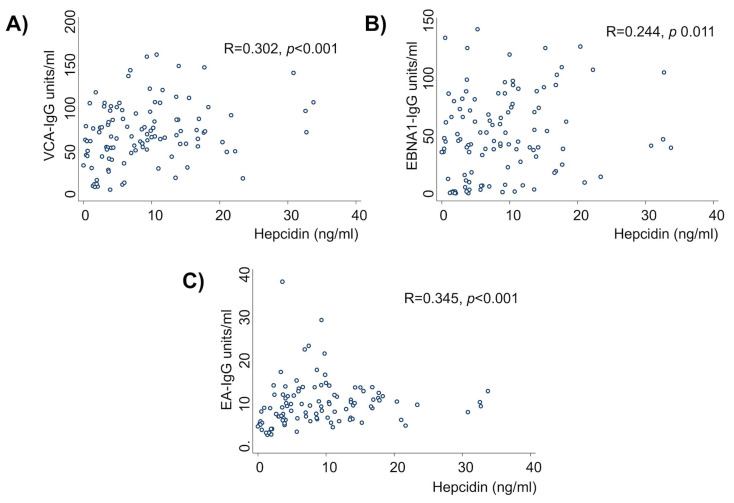
Hepcidin concentration positively correlates with the levels of EBV antibodies. (**A**) Viral capsid antigen (VCA), (**B**) EBV nuclear antigen 1 (EBNA1), and (**C**) early antigen (EA, also known as BMRF1) antibodies have a positive correlation with increased concentrations of hepcidin.

**Figure 3 ijms-24-01630-f003:**
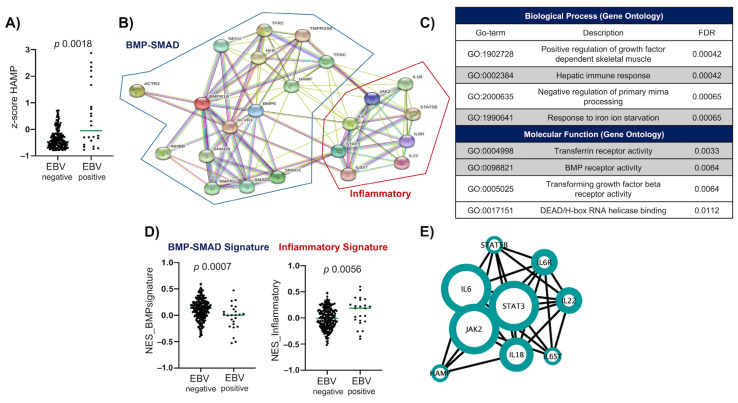
Elevated hepcidin (*HAMP)* expression is enriched in the EBV-associated gastric cancer samples (EBVaGC) through an inflammatory pathway. (**A**) *HAMP* expression in patients with EBVnGC (Epstein–Barr virus-negative gastric carcinoma) and EBVaGC. (**B**) Protein–protein interaction network (PPIN) of hepcidin and other proteins involved with hepcidin protein expression. (**C**) Gene Ontology (GO) analysis of biological processes and molecular functions related to the proteins of the PPIN. (**D**) Enrichment of BMP–SMAD and inflammatory signatures in EBVnGC and EBVaGC. (**E**) Centrality analysis where the size of the node represents the most important proteins in the PPIN, larger nodes indicate higher centrality. FDR = false discovery rate.

**Table 1 ijms-24-01630-t001:** Characteristics of study sample.

Characteristics	*N* (145)	%
Age, years, mean ± SD	9.3 ± 1.7	
Sex Girls Boys	70 75	48.3 51.7
Socio-economic level High margination ^a^ Middle or lower margination	86 56	60.6 39.4
Overcrowding <3 persons/room ≥3 persons/room	79 65	54.9 45.1
Body Mass Index-for-age (Z-score), mean ± SD ^b^	0.65 ± 1.02	
Thinness Normal BMI-for-age Overweight Obesity	1 95 33 16	0.7 65.5 22.8 11.0
Height-for-age (Z-score), mean ± SD ^c^	−0.41 ± 0.96	
Normal Stunting	137 8	94.5 5.5
Iron nutritional status and infectious/inflammation indicators		
Hemoglobin (g/dL), mean ± SD ^d^	14.2 ± 0.8	
Normal hemoglobin Anemia	144 1	99.3 0.7
Ferritin (ng/mL), median (p25–p75)	27.3 (19.6–41.7)	
Normal ferritin (≥15 ng/mL) Low iron stores (ferritin < 15 ng/mL)	122 23	84.1 15.9
Hepcidin (ng/mL), (median, p25–p75) *	7.6 (3.6–13.5)	
α-1 AGP (g/L) * median (p25–p75)	0.66 (0.51–0.83)	
Normal (<1 g/L) High (≥1 g/L)	93 13	87.7 12.3
C-reactive protein (mg/L) * median (p25–p75)	0.00 (0.00–1.84)	
Normal (<5 mg/L) High (≥5 mg/L)	91 14	86.7 13.3
Inflammatory process ^e^ Inflammatory process− Inflammatory process+	88 17	83.8 16.2

^a^ The first three quintiles in Socioeconomic level using Hollingshead index. ^b^ Body mass Index, Z-score (thinness < −2SD, normal −2SD ≤ Z-score < 1SD, overweight Z-score ≥ 1SD, obesity Z-score ≥ 2SD) [45]. ^c^ Height to age, Z-score (Normal −2SD ≤ Z-score < 3SD, Stunting Z-score < −2SD) [45]. ^d^ Hemoglobin (Hb) concentration (g/dL) normal Hb ≥ 11.5 g/dL for children less than 12 years, and Hb ≥ 12.0 g/dL for children aged 12 or more years. ^e^ C reactive protein > 5 mg/L and/or α-1 glycoprotein > 1 g/L. * With sufficient serum sample for determination: AGP ***N*** = 110, CRP ***N*** = 109, hepcidin ***N*** = 109.

**Table 2 ijms-24-01630-t002:** Frequency of infection by *H. pylori* and/or by Epstein–Barr Virus (EBV).

Infection Type	*N* = 145	
	*N*	%
*H. pylori* Positive (UBT+ or ELISA+) Negative (UBT− and ELISA−)	50 95	34.5 65.5
CagA pathogenicity island Positive Negative	29 111	20.7 79.3
EBV infection Positive Negative	133 12	91.7 8.3
*H. pylori* and EBV infection *H. pylori*+/EBV infection+ *H. pylori*+/EBV infection− *H. pylori*−/EBV infection+ *H. pylori*−/EBV infection−	47 3 86 9	32.4 2.1 59.1 6.2
With *H. pylori* or EBV infection or Inflammatory process Without *H. pylori,* EBV infection and Inflammatory process	135 7	95.1 4.9

**Table 3 ijms-24-01630-t003:** Acute-phase proteins concentration and infection with EBV or coinfection with EBV and *H. pylori*.

Infection	Ferritin (ng/mL)			C-Reactive Protein (mg/dL)	α-1 Glycoprotein (g/L)	Hepcidin (ng/mL)
	*N*	Median (p25–p75)	*N*	Median (p25–p75)		
EBV infection ^b^ With infection Without infection *p*-value ^a^	131 12	28.5 (19.4–42.1) 26.8 (21.7–32.1) 0.730	96 9	0.16 (0.00–2.34) 0.00 (0.00–0.00) 0.005	0.66 (0.55–0.88) 0.46 (0.33–0.54) 0.002	8.6 (3.7–13.6) 2.0 (1.7–3.9) 0.002
Coinfection *H. pylori*+/EBV infection+ ** H. pylori*+/EBV infection– *H. pylori*−/EBV infection+ *H. pylori*−/EBV infection– *p*-value ^b^	46 3 85 9	25.6 (20.1–37.9) 20.2 (5.8–26.5) 29.4 (19.4–45.9) 29.5 (26.0–34.1) 0.687	41 2 56 7	0.32 (0.00–2.30) 0.00 (0.00–0.00) 0.00 (0.00–3.10) 0.00 (0.00–0.00) 0.054	0.66 (0.53–0.84) 0.31 (0.30–0.32) 0.68 (0.55–0.89) 0.50 (0.43–0.68) 0.069	8.9 (4.2–13.5) ^c^ 1.5 (1.4–1.6) 8.6 (3.7–13.7) ^d^ 2.1 (1.9–5.7) ^e^ 0.030
With *H. pylori* or EBV infection or Inflammatory process Without *H. pylori* or EBV infection or Inflammatory process *p*-value ^a^	133 7	28.3 (19.4–41.7) 29.5 (26.0–34.1) 0.660	98 7	0.07 (0.00–2.33) 0.00 (0.00–0.00) 0.014	0.66 (0.53–0.88) 0.50 (0.43–0.68) 0.036	8.6 (3.7–13.6) 2.1 (1.9–5.7) 0.016

^a^ Mann–Whitney U test. ^b^ Kruskal–Wallis test Dunn with Bonferroni correction (^c^ vs. ^e^, *p* = 0.012; ^d^ vs. ^e^, *p* = 0.028) * This group was excluded from statistical test because it was very small.

## Data Availability

Not applicable.

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
