# Peer review of "Epstein–Barr Virus Infection Is Associated with Elevated Hepcidin Levels"

_ijms, 2023, doi:10.3390/ijms24021630_

Round 1

Reviewer 1 Report (Previous Reviewer 3)

Responses to original comments

1.  Your study   uses H Pylori as a basis of division in your analysis. It is important that H. Pylori is noted in the title. As a suggestion to the authors, one could state “Epstein-Barr Virus and Helicobacter pylori co-infection in children is associated with elevated serum Hepcidin. Remove or change chronic infection.

2. Although literature convention states EBV status as “EBV  (sero)positive individuals”, EBV carrier is an accepted term.

3. It is understandable that the limited number of individuals that comprise the purist data set, one should use caution when stating that the finding is significant. I would strongly suggest that the authors employ multivariate analysis when analyzing the data. Perhaps using EBV+/- and H pylori +/-  with the 3 markers values to further  strengthen the hypothesis.

“Figure 2 shows that all the markers measured of EBV chronic infection (VCA-IgG, 126 EA-IgG and EBNA1-IgG) had a positive and significant  correlation with the hepcidin concentration”. (Lines 126-127)

Please refrain form using the word “significant” when referring to the correlation between Herceptin and EBV serostatus. As stated in my previous review the R values being 30% or less are at best weak.

4. Clinically viewed most children acquire EBV infection when they are young (> 10) and when reached young adults are ~ 90% seropositive. Agreed these children have not “cleared” the infection rather they have resolved the acute infection.” In general, young children do not exhibit classical infectious mononucleosis, they are asymptomatic or exhibit mild “cold like” symptoms. Later these children are EBV positive but should be considered “healthy” meaning they have no infectious symptoms. Conversely common viral or bacterial infections children and adults’ results in acute innate responses  that leads to elevated interleukins (IL-6 driven) Herceptin as well as other acute phase proteins. PubMed 24478088  

 5. Although the authors hypothesis is intriguing that links EBV  and Herceptin. The one should be cautions in not ruling out other factors that can contribute or mimic the same outcome. It is well known that common viral and bacterial infections will increase Herceptin levels (reviewed in PubMed . 26291319). As the health status of the children were not know or stated in Table 1 or Methods section, adding a discussion on confounding factors is important as it allows the reader more assurance that the authors have considered other elements other than just EBV in their interpretation of the data.

 Additional comments:

Your Herceptin appear values  as values in healthy children are 21.89 ng/mL (16.50 to 51.70 ng/mL) in boys and 21.95 ng/mL (19.20 to 47.70 ng/mL) in girls. PubMed 29794646 As this may to a lack of precision in the data and could lead to a skewing of the values to measure only higher-level values. Please address this lower value issue in your Material and methods or Results section as a footnote in Table 1.

Author Response

Comments and Suggestions for Authors

Responses to original comments

  1. Your study uses H Pylori as a basis of division in your analysis. It is important that H. Pylori is noted in the title. As a suggestion to the authors, one could state “Epstein-Barr Virus and Helicobacter pylori co-infection in children is associated with elevated serum Hepcidin. Remove or change chronic infection.

Response. We have followed your advice and the title now reads “Epstein-Barr Virus and Helicobacter pylori Coinfection is Associated with Elevated Hepcidin Levels”.

  1. Although literature convention states EBV status as “EBV (sero)positive individuals”, EBV carrier is an accepted term.

Response. Thank you very much, we are maintaining the term EBV carrier throughout the text and title.

  1. It is understandable that the limited number of individuals that comprise the purist data set, one should use caution when stating that the finding is significant. I would strongly suggest that the authors employ multivariate analysis when analyzing the data. Perhaps using EBV+/- and H pylori +/- with the 3 markers values to further strengthen the hypothesis.

Response. We did test a multiple regression model to assess the association between EBV infection, H. pylori infection and coinfection with both pathogens and the hepcidin concentration (variable transformed as square root of hepcidin). We obtained similar results to those obtained with the univariate analysis, EBV infection was independently associated with hepcidin concentration, no association was observed between hepcidin and H. pylori infection, nor was any significant interaction or effect modification identified by the presence of H. pylori.

In our multivariate analysis, the only variable that proved to have a confounding effect was the concentration of AGP (AGP≥1 g/l vs. AGP<1 g/l). In the multiple linear regression model in which we evaluated the association between each pathogen or coinfection with hepcidin concentration after adjusting for AGP (high≥ 1g/l vs. normal < 1g/l), similar results were obtained, only EBV remained associated with hepcidin concentration (β coeficient 1.15, 95% CI [0.30 – 1.97], p=0.008). We considered that the multivariate analysis did not add new information to that already shown in Table 3, so we decided not to include it.

. regress sqrthepc1 i.EBVdi1 i.hpdico1 agp1di

     Source |       SS       df       MS             Number of obs =     105

-------------+------------------------------           F( 3,   101) =   3.07

       Model | 12.9907468     3 4.33024892           Prob > F     = 0.0311

   Residual | 142.329829   101 1.40920623           R-squared     = 0.0836

-------------+------------------------------           Adj R-squared = 0.0564

       Total | 155.320576   104 1.49346708           Root MSE     = 1.1871

------------------------------------------------------------------------------

   sqrthepc1 |     Coef.   Std. Err.     t   P>|t|     [95% Conf. Interval]

-------------+----------------------------------------------------------------

   1.EBVdi1 |   1.140882   .419992    2.72   0.008     .3077311   1.974034

   1.hpdico1 |   .1431312   .2382275     0.60   0.549     -.329448   .6157104

     agp1di |   .2444625   .3556137     0.69   0.493   -.4609794   .9499044

       _cons |   1.611281   .3992256     4.04   0.000    .8193246   2.403237

------------------------------------------------------------------------------

. regress sqrthepc1 i.EBVdi1## i.hpdico1 agp1di

     Source |       SS       df       MS             Number of obs =     105

-------------+------------------------------           F( 4,   100) =   2.43

       Model | 13.7632235     4 3.44080589           Prob > F     = 0.0525

   Residual | 141.557352   100 1.41557352           R-squared     = 0.0886

-------------+------------------------------           Adj R-squared = 0.0522

       Total | 155.320576   104 1.49346708           Root MSE     = 1.1898

--------------------------------------------------------------------------------

     sqrthepc1 |     Coef.   Std. Err.     t   P>|t|     [95% Conf. Interval]

---------------+----------------------------------------------------------------

     1.EBVdi1 |   .9689417   .4810052     2.01   0.047     .0146412   1.923242

     1.hpdico1 | -.5391316   .9539453   -0.57   0.573   -2.431732   1.353469

               |

EBVdi1#hpdico1 |

         1 1 |   .7278605   .9853072     0.74   0.462   -1.226961   2.682682

               |

       agp1di |   .2507591   .3565181     0.70   0.483   -.4565626   .9580808

         _cons |   1.762895   .4496941     3.92   0.000     .8707145   2.655075

--------------------------------------------------------------------------------

  1. “Figure 2 shows that all the markers measured of EBV chronic infection (VCA-IgG, 126 EA-IgG and EBNA1-IgG) had a positive and significant correlation with the hepcidin concentration”. (Lines 126-127). Please refrain form using the word “significant” when referring to the correlation between Herceptin and EBV serostatus. As stated in my previous review the R values being 30% or less are at best weak.

Response. We were using the word significant because the statistical tests showed a p value under 0.05. We agreed that the correlations are low and we included a comment about it in the limitations of the study. In the new version of the manuscript, we have eliminated the word significant, while we kept the p value and we discuss that it is a low correlation.

  1. Clinically viewed most children acquire EBV infection when they are young (> 10) and when reached young adults are ~ 90% seropositive. Agreed these children have not “cleared” the infection rather they have resolved the acute infection.” In general, young children do not exhibit classical infectious mononucleosis, they are asymptomatic or exhibit mild “cold like” symptoms. Later these children are EBV positive but should be considered “healthy” meaning they have no infectious symptoms. Conversely common viral or bacterial infections children and adults’ results in acute innate responses that leads to elevated interleukins (IL-6 driven) Herceptin as well as other acute phase proteins. PubMed 24478088

Response. We agree with this statement. We consider that the children included in the study were healthy at the time in which the serum samples were taken, as they did not present evidence of respiratory, urinary or gastric symptoms. They were EBV positive but did not shown evidence of infectious mononucleosis and they were not IgM positive. We comment on the limitations of the study that we cannot rule out additional infections responsible for the elevated hepcidin levels and of the other acute phase proteins.

  1. Although the authors hypothesis is intriguing that links EBV and Herceptin. The one should be cautions in not ruling out other factors that can contribute or mimic the same outcome. It is well known that common viral and bacterial infections will increase Herceptin levels (reviewed in PubMed . 26291319). As the health status of the children were not know or stated in Table 1 or Methods section, adding a discussion on confounding factors is important as it allows the reader more assurance that the authors have considered other elements other than just EBV in their interpretation of the data.

Response. We have complemented paragraph 2 of discussion in which we already have mentioned known viruses that regulate hepcidin and iron. In addition, we added a new paragraph in the discussion section (paragraph number 6) of the modified manuscript. In this new paragraph, we relate viral and bacterial infections with IL-1b and IL-6 as common regulators of the expression of acute phase proteins.

Additional comments:

Your Herceptin appear values as values in healthy children are 21.89 ng/mL (16.50 to 51.70 ng/mL) in boys and 21.95 ng/mL (19.20 to 47.70 ng/mL) in girls. PubMed 29794646 As this may to a lack of precision in the data and could lead to a skewing of the values to measure only higher-level values. Please address this lower value issue in your Material and methods or Results section as a footnote in Table 1.

Response. In the modified manuscript, we have complemented the information about the kit chosen to measure hepcidin levels (the human hepcidin-25 enzyme immunoassay kit from Bachem). The performance of this kit has been previously documented, comparing it with several other commercial tests (new references 95-97). This method is recommended by the International Consortium for Harmonization of Clinical Laboratory Results for the quantification of hepcidin. Also, the values obtained with this test are similar to others reported in children (see references 1-5 below). However, up to today, there is not a standard method for the quantification of serum hepcidin.

  1. Hyoung Soo Choi, Sang Hoon Song, Jae Hee Lee, Hee-Jin Kim, Hye Ran Yang. Serum hepcidin levels and iron parameters in children with iron deficiency. Korean J Hematol 2012.47:286-92. 10.5045/kjh.2012.47.4.286.
  2. Jadwiga Ambroszkiewicz, Witold Klemarczyk, Joanna Mazur, Joanna Gajewska, Grażyna Rowicka, MaÅ‚gorzata StruciÅ„ska, Magdalena CheÅ‚chowska. Serum Hepcidin and Soluble Transferrin Receptor in the Assessment of Iron Metabolism in Children on a Vegetarian Diet. Biol Trace Elem Res (2017) 180:182–190. 10.1007/s12011-017-1003-5.
  3. Mohammed Sanad, Mohammed Osman, Amal Gharib. Obesity modulate serum hepcidin and treatment outcome of iron deficiency anemia in children: A case control study. Italian Journal of Pediatrics 2011, 37:34.10.1186/1824-7288-37-34.
  4. Azab, SFA, Esh, AMH: Serum hepcidin levels in Helicobacter pylori-infected children with iron-deficiency anemia: a case-control study. Ann Hematol 2013, 92:1477-1483.10.1007/s00277-013-1813-2.
  5. Albertine E. Donker, Tessel E. Galesloot, Coby M. Laarakkers, Siem M. Klaver, Dirk L. Bakkeren, DorineW. Swinkels. Standardized serum hepcidin values in Dutch children: Set point relative to body iron changes during childhood. Pediatr Blood Cancer. 2020.67:e28038. 10.1002/pbc.28038.

Reviewer 2 Report (Previous Reviewer 2)

In this work, the authors have not examined the form of the EBV infection in the cohort of this study, in

 145 schoolchildren from boarding schools in Mexico City.

The authors defined the EBV serology status, determining by ELISA the levels of the EBV antibodies against the virus capsid antigen (VCA) (IgM anti-VCA and IgG anti-VCA), virus early antigen (EA) (IgG anti-EA), and virus nuclear antigen 1 (EBNA1) (IgG anti-EBNA1), in the serum of children in this study.

The presence of the EBV has not been analyzed,- not the presence of the EBV DNA, or EBV RNA, or EBV proteins. Even more, the all of the children included in the study were IgM anti-VCA negative. It means that 133 children, which were positive for the IgG anti-VCA and negative for the IgM anti-VCA, had EBV past infection [see in: Massimo De Paschale, Pierangelo Clerici. Serological diagnosis of Epstein-Barr virus infection: Problems and solutions. World J Virol 2012 February 12; 1(1): 31-43. doi: 10.5501/wjv.v1.i1.31.].

The EBV chronic infection is not proved in this study.

 Consequently:

1. The authors must replace the wrong definition “EBV chronic infection”, as well as “chronic EBV infection” with the correct one - “EBV infection”, all over in the text of the manuscript, including the Title and the Abstract.

 2. The EBV is the main player in this report. This virus is known and is studied for the last 60 years.

I suggest for authors to include into Introduction/Discussion the information about EBV, the information that is related to their study.

See the examples in the published articles, which you have referred to:

1)     Abigail Morales-Sánchez et al., Detection of Epstein-Barr Virus DNA in Gastric Biopsies of Pediatric Patients with Dyspepsia. Pathogens 2020, 9, 623; doi:10.3390/pathogens9080623

In the Discussion:

“EBV infects close to 95% of the adult population worldwide, with memory B cells being the main reservoir of viral latent infection. There is experimental and clinical evidence of the capacity of EBV to also infect epithelial cells, …..[1012].”

“Elevated anti-VCA antibodies mark individuals with higher risk to develop nasopharyngeal carcinoma [3032], supporting a mechanism of enhanced EBV reactivation from latently infected B cells favoring enhanced infection of the nasopharyngeal epithelial cells. Others have also made a similar observation for EBV and GC [4,33,34].”

2)     Cárdenas-Mondragón, M.G. et al., Epstein Barr Virus and Helicobacter Pylori Co-Infection Are Positively Associated with Severe Gastritis in Pediatric Patients. PLoS ONE 2013, 8, e62850, doi:10.1371/journal.pone.0062850.

In the Introduction: “EBV infection has been consistently associated with several types of lymphoma, nasopharyngeal carcinoma (NPC) [10], [11] and more recently to GC [12], [13], [14]. EBV infection also occurs early in childhood and usually persists in B cells, with most infected individuals carrying the virus asymptomatically in a latent stage in these cells. It is not clear when EBV infects the gastric mucosa and whether infection induces an inflammatory reaction, as observed with H. pylori. EBV reactivation from infected B cells has been proposed to facilitate infection of the epithelial basolateral face [15]. In that scenario, the titer of anti-EBV antibodies against structural proteins has been proposed to correlate with the level of viral reactivation and as a prognostic marker in NPC [10], [16], [17].”

3)     See in the Review by Claire Shannon-Lowe and Alan Rickinson. The Global Landscape of EBV-Associated Tumors. Front. Oncol. 9:713. doi: 10.3389/fonc.2019.00713,

in the chapter THE BIOLOGY OF EBV INFECTION: AN OVERVIEW:

“…during EBV lifelong persistence within the B cell system,…. the concept of a Latency 0 infection (expressing only the EBERs and BART-miRs) is directly inferred from the fact that non-proliferating, EBER/BART-miR-positive but virus antigen-negative, B cells constitute the major reservoir of latent infection in the blood of healthy virus carriers (18). ….

these circulating Latency 0 cells lie exclusively within the memory B cell pool (30)…..

plasma cell differentiation is known to trigger reactivation from latency into lytic cycle (48). Virus-producing plasmablasts could initiate new infections of neighboring B cells, potentially topping up the latent B cell reservoir, or if situated at a sub-epithelial site as has been seen in tonsillar crypts (12) could seed new epithelial foci of virus replication, …..”

Author Response

Comments and Suggestions for Authors

In this work, the authors have not examined the form of the EBV infection in the cohort of this study, in

145 schoolchildren from boarding schools in Mexico City.

The authors defined the EBV serology status, determining by ELISA the levels of the EBV antibodies against the virus capsid antigen (VCA) (IgM anti-VCA and IgG anti-VCA), virus early antigen (EA) (IgG anti-EA), and virus nuclear antigen 1 (EBNA1) (IgG anti-EBNA1), in the serum of children in this study.

The presence of the EBV has not been analyzed,- not the presence of the EBV DNA, or EBV RNA, or EBV proteins. Even more, the all of the children included in the study were IgM anti-VCA negative. It means that 133 children, which were positive for the IgG anti-VCA and negative for the IgM anti-VCA, had EBV past infection [see in: Massimo De Paschale, Pierangelo Clerici. Serological diagnosis of Epstein-Barr virus infection: Problems and solutions. World J Virol 2012 February 12; 1(1): 31-43. doi: 10.5501/wjv.v1.i1.31.].

The EBV chronic infection is not proved in this study.

 Consequently:

  1. The authors must replace the wrong definition “EBV chronic infection”, as well as “chronic EBV infection” with the correct one - “EBV infection”, all over in the text of the manuscript, including the Title and the Abstract.

Response. We have removed the word chronic from the text and we are using EBV carriers instead. We were using the word chronic to distinguish from a primary infection since all the children were IgM negative. We have never wanted to convey that the children had symptoms associated with EBV infection or had detectable virus.

  1. The EBV is the main player in this report. This virus is known and is studied for the last 60 years. I suggest for authors to include into Introduction/Discussion the information about EBV, the information that is related to their study.

See the examples in the published articles, which you have referred to:

1)     Abigail Morales-Sánchez et al., Detection of Epstein-Barr Virus DNA in Gastric Biopsies of Pediatric Patients with Dyspepsia. Pathogens 2020, 9, 623; doi:10.3390/pathogens9080623

In the Discussion:

“EBV infects close to 95% of the adult population worldwide, with memory B cells being the main reservoir of viral latent infection. There is experimental and clinical evidence of the capacity of EBV to also infect epithelial cells, …..[10–12].”

“Elevated anti-VCA antibodies mark individuals with higher risk to develop nasopharyngeal carcinoma [30–32], supporting a mechanism of enhanced EBV reactivation from latently infected B cells favoring enhanced infection of the nasopharyngeal epithelial cells. Others have also made a similar observation for EBV and GC [4,33,34].”

2)     Cárdenas-Mondragón, M.G. et al., Epstein Barr Virus and Helicobacter Pylori Co-Infection Are Positively Associated with Severe Gastritis in Pediatric Patients. PLoS ONE 2013, 8, e62850, doi:10.1371/journal.pone.0062850.

In the Introduction: “EBV infection has been consistently associated with several types of lymphoma, nasopharyngeal carcinoma (NPC) [10], [11] and more recently to GC [12], [13], [14]. EBV infection also occurs early in childhood and usually persists in B cells, with most infected individuals carrying the virus asymptomatically in a latent stage in these cells. It is not clear when EBV infects the gastric mucosa and whether infection induces an inflammatory reaction, as observed with H. pylori. EBV reactivation from infected B cells has been proposed to facilitate infection of the epithelial basolateral face [15]. In that scenario, the titer of anti-EBV antibodies against structural proteins has been proposed to correlate with the level of viral reactivation and as a prognostic marker in NPC [10], [16], [17].”

3)     See in the Review by Claire Shannon-Lowe and Alan Rickinson. The Global Landscape of EBV-Associated Tumors. Front. Oncol. 9:713. doi: 10.3389/fonc.2019.00713,

in the chapter THE BIOLOGY OF EBV INFECTION: AN OVERVIEW:

“…during EBV lifelong persistence within the B cell system,…. the concept of a Latency 0 infection (expressing only the EBERs and BART-miRs) is directly inferred from the fact that non-proliferating, EBER/BART-miR-positive but virus antigen-negative, B cells constitute the major reservoir of latent infection in the blood of healthy virus carriers (18). ….

these circulating Latency 0 cells lie exclusively within the memory B cell pool (30)…..plasma cell differentiation is known to trigger reactivation from latency into lytic cycle (48). Virus-producing plasmablasts could initiate new infections of neighboring B cells, potentially topping up the latent B cell reservoir, or if situated at a sub-epithelial site as has been seen in tonsillar crypts (12) could seed new epithelial foci of virus replication, …..”

Response. Following your advice, we have extended the introductory section related to EBV and relevant to this study. We have added a new paragraph (2nd) and we have more carefully depicted the combined role of H. pylori and EBV on gastric lesions (5th paragraph).

Reviewer 3 Report (New Reviewer)

Duque et al. studied the levels of selected acute phase proteins of in sera collected from 131 EBV positive and 12 EBV negative children. A subset of EBV positive children as well as a fraction of EBV negative children was infected with Helicobacter pylori, a bacterium associated with iron deficiency and iron deficiency anemia. Although there was no significant diifference in the serum level of ferritin (a potential biomarker of iron status), between EBV-infected and uninfected children, the serum levels of hepcidin, C-reactive protein (CRP) and α-1 glycoprotein (AGP) were significantly higher in the EBV positive group. Chronic EBV infection per se was associated with elevated hepcidin and AGP levels, whereas an increase in serum CRP was characteristic for children coinfected with EBV and Helicobacter pylori. Serum hepcidin levels correlated with the levels of IgG antibodies directed against the EBV proteins VCA, EBV and EA. Using data deposited to TCGA, the authors also analysed the expression of certain genes encoding distinct acute phase proteins in EBV-positive and EBV-negative gasreic carcinomas. They found an elevated expression of the hepcidin-encoding HAMP gene in EBV-positive gastric carcinomas and based on a protein-protein interaction network of hepcidin and its potential regulators, followed by gene ontology analysis and single sample gene enrichment analysis they suggested a role foe IL-6, IL-1β, JAK2 and STAT3 in the upregulation of hepcidin expression in EBV-associated gastric carcinomas.

This is an interesting study and I recommend the publication of the manuscript after a minor revision. I suggest the following changes:

line 20, …TCGA cancer gastric database… change to: …TCGA gastric cancer database…

line 27 and 156, Helicobacter pylori – please use italics

lines 37 and 38, H. pylori has the stomach as the main site of infection.…- please amend that sentence

line 138 (Figure 2, title), …significant… change to: significantly

line 157, …HAMP…– please use italics

line 157, …A1BG…– please use italics

line 157, …CRP…– please use italics

line 189, …HAMP… change to …HAMP…  -please notice that it denotes the protein in this case

line 224 and 475, …school children… change to: schoolchildren

line 245, …occur…- please use another expression

line 374, …HU units/ml – please specify the meaning of HU or please skip it

line 636, …CORREA, ….PIAZUELO, please change to: …Correa,…Piazuelo,…

I recommend the publication of the manuscript after a minor revision.

Author Response

Duque et al. studied the levels of selected acute phase proteins of in sera collected from 131 EBV positive and 12 EBV negative children. A subset of EBV positive children as well as a fraction of EBV negative children was infected with Helicobacter pylori, a bacterium associated with iron deficiency and iron deficiency anemia. Although there was no significant diifference in the serum level of ferritin (a potential biomarker of iron status), between EBV-infected and uninfected children, the serum levels of hepcidin, C-reactive protein (CRP) and α-1 glycoprotein (AGP) were significantly higher in the EBV positive group. Chronic EBV infection per se was associated with elevated hepcidin and AGP levels, whereas an increase in serum CRP was characteristic for children coinfected with EBV and Helicobacter pylori. Serum hepcidin levels correlated with the levels of IgG antibodies directed against the EBV proteins VCA, EBV and EA. Using data deposited to TCGA, the authors also analysed the expression of certain genes encoding distinct acute phase proteins in EBV-positive and EBV-negative gasreic carcinomas. They found an elevated expression of the hepcidin-encoding HAMP gene in EBV-positive gastric carcinomas and based on a protein-protein interaction network of hepcidin and its potential regulators, followed by gene ontology analysis and single sample gene enrichment analysis they suggested a role foe IL-6, IL-1β, JAK2 and STAT3 in the upregulation of hepcidin expression in EBV-associated gastric carcinomas.

This is an interesting study and I recommend the publication of the manuscript after a minor revision. I suggest the following changes:

line 20, …TCGA cancer gastric database… change to: …TCGA gastric cancer database…

line 27 and 156, Helicobacter pylori – please use italics

lines 37 and 38, H. pylori has the stomach as the main site of infection.…- please amend that sentence

line 138 (Figure 2, title), …significant… change to: significantly

line 157, …HAMP…– please use italics

line 157, …A1BG…– please use italics

line 157, …CRP…– please use italics

line 189, …HAMP… change to …HAMP… -please notice that it denotes the protein in this case

line 224 and 475, …school children… change to: schoolchildren

line 245, …occur…- please use another expression

line 374, …HU units/ml – please specify the meaning of HU or please skip it

line 636, …CORREA, ….PIAZUELO, please change to: …Correa,…Piazuelo,…

I recommend the publication of the manuscript after a minor revision.

Response. Thank you very much for your positive comments and an apology for the typos. We have made all the corrections to the text. We have double-checked and now italics are only used when referring to the gene, and italics are not used when referring to the protein. 

Reviewer 4 Report (New Reviewer)

H. pylori and EBV infections are highly common in children. EBV is well known an etiological agent of several cancers e.g. gastric cancer. H. pyloriplay also a significant role in the gastric cancer development.  

The topic is interesting; the methodology is correct. The study group raises doubts. 

Major comments: 

1.     The purpose of the research was not clearly defined.

Line 74-77  - The main goal of this study was to evaluate a potential link between EBV and hepcidin and the acute phase proteins CRP and AGP. 

2.     What patient groups were studied 

3.     On what basis the authors compared a group of children with patients with EBVaGC

Line 320-322 – “The study  involved children 6 to 13 years of age, who initially participated in a longitudinal study about the effect of H. pylori infection on growth velocity in schoolchildren with low socioeconomical status”. 

Line 223-224 -  “In this study, we documented a significant association between EBV infection and elevated levels of hepcidin, both in school children and in adults with EBVaGC”. 

This must be clearly explained in the text. In this form, it is incomprehensible to the reader. 

Author Response

Comments and Suggestions for Authors

  1. pyloriand EBV infections are highly common in children. EBV is well known an etiological agent of several cancers e.g. gastric cancer. H. pyloriplay also a significant role in the gastric cancer development.  

The topic is interesting; the methodology is correct. The study group raises doubts. 

Major comments: 

  1. The purpose of the research was not clearly defined.

Line 74-77  - The main goal of this study was to evaluate a potential link between EBV and hepcidin and the acute phase proteins CRP and AGP. 

Response. Since H. pylori has been associated with upregulation of hepcidin and other acute phase proteins and different lines of evidence support interactions and even synergy between H. pylori and EBV during infection-induced damage to the gastric mucosa, we wanted to assess whether EBV was also involved in the regulation of these metabolites. The study was planned with the schoolchildren cohort, but of about 350 samples collected, we only had about 150 samples for all the tests included in this study. Unfortunately, we think that this number of children in the study prevented us to also observe associations with the children´s nutritional and anthropometric measures. That is why we decided to analyze the TCGA adult cohort as a proof of concept that EBV was upregulating hepcidin (HAMP) levels and/or acute phase proteins. Because we made similar observations in this cohort, an association with hepcidin and with the hepcidin regulatory pathway through inflammation, we decided to add this data to the study. We have made modifications to the abstract and to the last paragraph of the introductory section to better convey the use of both children and adult cohorts in the study. We hope that with these modifications the study does not feel disconnected and both analyzes read more congruent.

  1. What patient groups were studied 

Response. We now explained at the end of the introductory section that two groups are studied. This section now reds as follows “We studies two cohorts for this analysis, children from boarding schools in Mexico City and adults with gastric cancer from The Cancer Genome Atlas (TCGA) consortium. The former allowed us to also analyze the nutritional status of children, while the latter confirmed the observations made in children plus allowed us to look for molecular mechanisms of EBV and hepcidin interaction”.

  1. On what basis the authors compared a group of children with patients with EBVaGC

Line 320-322 – “The study  involved children 6 to 13 years of age, who initially participated in a longitudinal study about the effect of H. pylori infection on growth velocity in schoolchildren with low socioeconomical status”. Line 223-224 -  “In this study, we documented a significant association between EBV infection and elevated levels of hepcidin, both in school children and in adults with EBVaGC”. 

This must be clearly explained in the text. In this form, it is incomprehensible to the reader. 

Response. We decided to also assess the observations made in children in a cohort of adult patients with gastric cancer because the co-infection of H. pylori and EBV has been associated with the progressive damage of the gastric mucosa and the development of gastric cancer. One of the mechanisms proposed by which H. pylori alters the levels of hepcidin is through erosion of the gastric mucosa, therefore, EBV could be having a similar effect. We have extended the last paragraph of the introductory section to better convey the point that EBV could be altering the gastric mucosa similar to and together with H. pylori. This section now reads as follows “Multiple lines of evidence support interactions and synergy between H. pylori and EBV infection in the gastric mucosa [14,33,36–42]. These studies support that H. pylori facilitates the arrival of EBV infected B lymphocytes to the stomach [42], EBV reactivation and epithelial cell infection [40,41,43], triggering of severe lesions in coinfected individuals compared with those infected by only one pathogen [14,33,36,44], and synergism in their oncogenic activity [38]. Whether EBV infection also influences the levels of proteins related to iron metabolism, specifically hepcidin, has never been reported”.

Round 2

Reviewer 2 Report (Previous Reviewer 2)

The results presented in the manuscript are interesting and are deserved to be published.

I agree with the authors that although the results of this study does not allow to conclude that it is EBV the driving force activating the inflammatory pathway and the enhanced expression of hepcidin, it invites for further research on this topic, placing the attention on EBV infection as a potential element influencing iron metabolism and anemia in children and adults.

The minor revision is needed.

I. Two imprecisions in Conclusion do not correspond exactly to the results of this study and should be clarified:

1.) The 1st sentence: “The results of the study support that children carriers of EBV infection present high levels of hepcidin and other acute phase proteins, such as CRP and AGP.”

However, the results of this study has demonstrated that “Hepcidin and AGP remained high in children solely infected with EBV, while CRP was only significantly high in coinfected children.” (Abstract lines 7-8 and Table 3).

Suggestion: to include in Conclusion after the 1st sentence the precision (similar to those in the Abstract).

2.) The 3d sentence: “The EBV influence on hepcidin expression may be a side effect of the virus-induced inflammation, particularly at gastric, gastro-intestinal or hepatic tissue, since we also observed a link between IL-6 and IL-1β and EBVaGC,…”

However, there are no results in this study regarding the gastro-intestinal or hepatic tissues, and EBVaGC, it is the EBV associated gastric cancer patients,

Suggestion: to remove or to precise.

 II. I also suggest the following minor corrections:

1.) in Abstract, line 6: “We found that children IgG positive to EBV infection…”, to precise to  “We found that children IgG positive to EBV antigens (VCA, EBNA1, and EA)…”;

2.) in Figure 2 legend:Hepcidin concentration positively correlates with EBV antibodies.” to precise to  “Hepcidin concentration positively correlates with the levels of EBV antibodies.”;

3.) page 8, paragraph 2, line 9: “a correlation between EBV infection and the concentration of hepcidin…” to correct to “a correlation between EBV infection and the expression of hepcidin…”;

4.) page 11, paragraph 2, lines 6-7: “against the latent gene EBNA1.” to correct to “against the latent protein EBNA1.”

Author Response

The results presented in the manuscript are interesting and are deserved to be published.

I agree with the authors that although the results of this study does not allow to conclude that it is EBV the driving force activating the inflammatory pathway and the enhanced expression of hepcidin, it invites for further research on this topic, placing the attention on EBV infection as a potential element influencing iron metabolism and anemia in children and adults.

The minor revision is needed.

  1. Two imprecisions in Conclusion do not correspond exactly to the results of this study and should be clarified:

1.) The 1st sentence: “The results of the study support that children carriers of EBV infection present high levels of hepcidin and other acute phase proteins, such as CRP and AGP.”

However, the results of this study has demonstrated that “Hepcidin and AGP remained high in children solely infected with EBV, while CRP was only significantly high in coinfected children.” (Abstract lines 7-8 and Table 3).

Suggestion: to include in Conclusion after the 1st sentence the precision (similar to those in the Abstract).

Response. According to the reviewer´s suggestion the conclusion now reads as follows: The results of the study support that children carriers of EBV infection present high levels of hepcidin and other acute phase proteins, such as CRP and AGP. Hepcidin and AGP remained high in children solely infected with EBV, while CRP was only significantly high in EBV and H. pylori coinfected children.

2.) The 3d sentence: “The EBV influence on hepcidin expression may be a side effect of the virus-induced inflammation, particularly at gastric, gastro-intestinal or hepatic tissue, since we also observed a link between IL-6 and IL-1β and EBVaGC,…”

However, there are no results in this study regarding the gastro-intestinal or hepatic tissues, and EBVaGC, it is the EBV associated gastric cancer patients,

Suggestion: to remove or to precise.

Response. We have eliminated the link with the gastro-intestinal or hepatic tissues. The sentence now reads. The sentence now reads: The EBV influence on hepcidin expression may be a side effect of the virus-induced inflammation, since we also observed a link between IL-6 and IL-1β and EBVaGC.

  1. I also suggest the following minor corrections:
  • in Abstract, line 6: “We found that children IgG positive to EBV infection…”, to precise to “We found that children IgG positive to EBV antigens (VCA, EBNA1, and EA)…”;

Response. We have made the suggested change.

  • in Figure 2 legend:Hepcidin concentration positively correlates with EBV antibodies.” to precise to “Hepcidin concentration positively correlates with the levels of EBV antibodies.”;

Response. We have made the suggested change.

  • page 8, paragraph 2, line 9: “a correlation between EBV infection and the concentration of hepcidin…” to correct to “a correlation between EBV infection and the expression of hepcidin…”;

Response. We have made the suggested change.

4.) page 11, paragraph 2, lines 6-7: “against the latent gene EBNA1.” to correct to “against the latent protein EBNA1.”

Response. We have made the suggested change.

Reviewer 4 Report (New Reviewer)

This paper has been corrected according all comments.

Author Response

This paper has been corrected according all comments.

Response. Thank you very much.

This manuscript is a resubmission of an earlier submission. The following is a list of the peer review reports and author responses from that submission.

Round 1

Reviewer 1 Report

The authors aim to study the relationship between EBV infection alone or in confection with H. pylori  with Hepcidin and acute phase proteins (CRP and AGP). They did observe and association between EBV infection and Hepcidin, however there are some concerns:

1) The paper is hard to follow. Table 2 and Table 3 are not referenced in the text, even though the data is reference in the text. This makes the paper very hard to follow and confusing. 

2) It is not explained what is EBV aGC and EBV nGC, this is a main part of the paper and it is not described what it stands for therefore it is hard to understand what the results mean. 

3) All the conclusions are made by associations, there is no direct evidence in the paper on their claims. 

1) The authors observe an association between EBV infection and Hepcidin, however there is no association with iron homeostasis. How do they explain this since Hepcidin is associated with iron homeostasis?

Reviewer 2 Report

The results presented in the manuscript are interesting and are deserved to be published after the revision.

Questions I put regarding EBV must be resolved.

Reviewer 3 Report

The title should be changed to reflect the input of Helicobacter pylori

Given the different types of EBV infection reported in the literature, it is misleading to categorize this pediatric cohort as chronically infected. From the literature chronic EBV infection is characterized as individuals who demonstrate high levels of productive (lytic) EBV infection. Children of the authors cohort who experienced primary  infection at an earlier time would be reported as latently infected. Although two individuals appeared to show acute (productive infection) as evidence by EBV IgM serology status, they were removed from the tabulations.

Number of EBV negative and Helicobacter pylori (HP) positive children is only 3 (Table 2). Findings from just 3 individuals which could be greatly impacted is even one was eliminated. The grouping is perhaps too low to draw any statistically significant conclusions for data presented in Table 3.

This reviewer considers their hypothesis based on earlier EBV infection (IgG positive and healthy status) impacts hepcidin in HP+ children a bit of a “stretch”. The children’s initial viral infection would have occurred at a minimum of 6 months or more earlier and is now resolved the infection.  Given the apparent close quarters, sanitary or nutritional conditions at the school (as seen by a ponderance HP infection and obesity levels), it would not be surprising that other common viral or bacterial infections occurred that would also increase IL-6/IL-1, acute phase proteins and Hepcidin. Without the health status of the child during the times shortly proceeding or at blood draw, one cannot easily link EBV and hepcidin or draw any solid conclusions to support the hypothesis.  

As a rule, in population studies R values of less than 50<R>20 are moderately positive while R values x<20 as weakly positive. As noted in Figure 2 all R values are at best weak or weak-moderate positive. The authors should use caution when arguing any strong association between cohort EBV IgG status and hepcidin as significant.

When correlating EBV, inflammatory signals or markers such as hepcidin to gastric cancer, the authors should address 1or compare  other confounding factors commonly seen with gastric cancer. These include HP or general changes in hepatic physiology during cancer  that would easily provide equally alternative explanations for elevated hepcidin.   

Discussion section:

This reviewer agrees that lytic infection as seen during acute infectious mononucleosis increased the levels of proinflammatory IL-6 and acute phase proteins. However, after virus resolution, IL-6 levels return to that of healthy (uninfected) individuals.  Discussion of altered liver/iron metabolism and the use of HCV and HIV as viral comparisons is misleading. Both viruses have significant virus replication as seen by their high viral loads. This compares with EBV seropositive individuals who, after initial infection show extremely low and intermittent occurrences of reactivation (lytic) as measured by serology (IgM) or viral load (PCR of peripheral blood or throat washings).